# A Perspective on Therapeutic Pan-Resistance in Metastatic Cancer

**DOI:** 10.3390/ijms21197304

**Published:** 2020-10-03

**Authors:** Dimitrios Korentzelos, Amanda M. Clark, Alan Wells

**Affiliations:** 1Department of Pathology, University of Pittsburgh, Pittsburgh, PA 15261, USA; Korentzelosd@upmc.edu (D.K.); amc235@pitt.edu (A.C.); 2UPMC Hillman Cancer Center, University of Pittsburgh, Pittsburgh, PA 15213, USA; 3VA Pittsburgh Healthcare System, Pittsburgh, PA 15213, USA; 4Department of Bioengineering, University of Pittsburgh, Pittsburgh, PA 15260, USA; 5Department of Computational & Systems Biology, University of Pittsburgh, Pittsburgh, PA 15260, USA

**Keywords:** metastasis, disseminated tumor cells, e-cadherin, therapy resistance, dormancy, metastatic microenvironment, immune checkpoint blockade, epigenetics, metabolic plasticity

## Abstract

Metastatic spread represents the leading cause of disease-related mortality among cancer patients. Many cancer patients suffer from metastatic relapse years or even decades after radical surgery for the primary tumor. This clinical phenomenon is explained by the early dissemination of cancer cells followed by a long period of dormancy. Although dormancy could be viewed as a window of opportunity for therapeutic interventions, dormant disseminated cancer cells and micrometastases, as well as emergent outgrowing macrometastases, exhibit a generalized, innate resistance to chemotherapy and even immunotherapy. This therapeutic pan-resistance, on top of other adaptive responses to targeted agents such as acquired mutations and lineage plasticity, underpins the current difficulties in eradicating cancer. In the present review, we attempt to provide a framework to understand the underlying biology of this major issue.

## 1. Introduction

Despite the major advances in cancer treatment in recent decades, most of the progress has been achieved through early diagnosis and treatment of pre-metastatic cancer. Unfortunately, metastatic disease remains essentially incurable for most cancers, as metastases are not amenable to removal due to wide dissemination, exhibit intrinsic generalized resistance to chemotherapy and immunotherapy, and rapidly develop acquired resistance to targeted therapy via adaptive mutations [1]. It is imperative to understand the biology of this generalized resistance in order to translate this understanding into novel therapeutic approaches that will improve patient outcomes.

The generalized resistance of metastases to chemotherapy is most clearly seen with neoadjuvant breast cancer treatment (systemically administered drugs before breast cancer surgery). Despite the fact that neoadjuvant chemotherapy will shrink the breast tumor, allowing a breast-conserving surgery and oftentimes leading to a complete pathological response, this fails to reflect on clinical outcomes, such as event-free survival and overall survival [2,3,4,5]. Thus, disseminated tumor cells (DTCs) that have left the primary tumor before resection often appear not to be eradicated by therapy but instead are intrinsically resistant. This differential therapeutic responsiveness between metastases and primary tumors may be attributed to acquired mutations in some cases of early dissemination or may be microenvironmentally dictated in cases of late dissemination, where primary tumors and their metastases are genetically closely related [6,7,8]. The question regarding the temporal occurrence of metastasis is a controversial one, and most likely, different types of cancer display different progression trajectories to systemic disease with the microenvironment being a central player either via direct effects to DTCs or indirectly by providing the optimal conditions for acquisition of genetic changes.

This review aims to discuss the phenomenon of therapeutic pan-resistance of DTCs, micrometastases, and macrometastases. We will focus on the concept of cellular dormancy and its implications for resistance to chemotherapy and immunotherapy, as well as the role of the dynamic crosstalk of the tumor with the metastatic microenvironment (MME) and the immune system. Although significant uncertainties remain, a plethora of recent studies aided by novel experimental platforms have provided significant insight into the mechanisms that allow DTCs and micrometastases to resist chemotherapy and immunotherapy during dormancy and then outgrow into lethal macrometastases. 

## 2. The Invasion-Metastasis Cascade

From an evolutionary perspective, metastasis can be thought of as a linear sequence of events, collectively described in the literature as the invasion-metastasis cascade [9]. In order for the cancer cells to arrive at the site of metastasis, they have to undergo a series of adaptations, including local invasion, intravasation, bloodborne dissemination, extravasation, and colonization, as well as coping with foreign environments much different from their tissue of origin [10]. At this point, cells that arrive in the metastatic setting exist either as DTCs or micrometastases [11]. While DTCs are solitary, dormant cells in a truly quiescent state, micrometastases most likely exist in a state of punctuated quiescence where proliferation is not continuous but rather sporadic before being suppressed, in contrast to previous assumptions in favor of a continuous balanced equilibrium between cell division and apoptosis [12]. It has to be emphasized that the aforementioned adaptations are based on stochastic events, and consequently, there is a high attrition rate of cells in hostile environments rendering metastasis an inefficient process [13,14]. Eventually, secondary to certain local or systemic events, DTCs or micrometastatic deposits exit from dormancy and start proliferating, giving rise to the actively growing, vascularized, lethal macrometastases [15]. 

In this context, dormancy and reawakening provide a solid explanation for the long periods of apparent stability seen in many cases of cancer, including breast cancer, prostate cancer, and melanoma [10]. Strikingly, excess mortality in breast cancer patients can be documented up to 20 years after surgery [16], while circulating breast cancer cells have been detected in patients clinically free of disease up to 22 years after diagnosis [17]. An interesting observation is that patients with HER2+ or triple negative (TN) breast cancer tend to relapse early, within five years from surgery, while ER+ cancers show a relatively stable rate of relapse over a period of several years [18,19]. In a similar fashion, in prostate cancer the median time from biochemical only recurrence (defined as an increase in prostate-specific antigen; PSA) after radical prostatectomy to bone metastasis and death exceed 16 years [20]. Such late and ultra-late recurrences are frequently seen in other neoplasms, such as melanoma and renal cell carcinoma [21,22], and illustrate the sequelae of the invasion-metastasis cascade as well as highlight the difficulty of efficiently treating DTCs and micrometastases. 

## 3. Cancer-Associated Epithelial-Mesenchymal Plasticity

Epithelial-mesenchymal transition (EMT) is a biologic process with established roles in numerous developmental programs involved in new tissue and organ generation and is usually followed by the reverse process, mesenchymal-epithelial transition (MET) [23,24,25]. Cancer cells exploit these processes, namely cancer-associated epithelial-mesenchymal transition (cEMT) and cancer-associated mesenchymal-epithelial reverting transition (cMErT) to obtain a number of the classical hallmarks of cancer [26,27], mainly increased metastatic potential and enhancement of the cancer stem cell (CSC) phenotype [28,29,30]. It has to be emphasized that at any given time point cancer cells are not fixed either in a purely epithelial or mesenchymal state, but rather at some point along the epithelial-mesenchymal axis, harnessing beneficial properties from both polar states [31]. As a result, many scientists prefer the term cancer-associated epithelial-mesenchymal plasticity (cEMP) underpinning the hybrid state of these cells and the bidirectional potential based on internal and external cues. 

The foundation for the role of cEMP in metastasis arises from observations and functional evidence of the increased ability of mesenchymally shifted carcinoma cells to escape from the primary tumor, along with acquisition of increased survival, stemness, and metastasis-initiating capacity compared to cancer cells with epithelial characteristics [23]. Nevertheless, the fact that induction of stable mesenchymal features abrogates metastatic outgrowth in preclinical models, together with clinical observations that metastases exhibit similar (or even enhanced) epithelial properties relative to their primary tumors, complicates the importance of cEMP in metastasis [28,32,33]. cEMP is also linked to stemness of tumor cells, which confers features supportive of successful outgrowth of metastatic deposits. Consistently, breast cancer cells that present a CSC phenotype frequently exhibit cEMT, and the abrogation of this process impinges on their stemness [34,35,36,37]. This relationship is not that straightforward though, as studies have shown an uncoupling of stemness from the mesenchymal state, and more specifically knockdown of the cEMT transducer PRRX1 reversed cEMT and promoted metastasis in breast cancer cells, with the carcinoma derived from this cMErT retaining stemness properties [38].

## 4. Role of E-Cadherin: Friend or Foe?

While cEMT is considered essential for successful tumor cell migration and invasion, it is now well accepted that cMErT confers increased survival and aids in successful colonization of metastatic sites by cancer cells [1]. Within this context, E-cadherin, a calcium-dependent cell-cell adhesion protein traditionally viewed as a tumor suppressor, seems to have a powerful role in therapeutic pan-resistance of micrometastases.

Early evidence for the importance of cMErT with regards to metastatic outgrowths came from observations of equal or increased levels of E-cadherin expression in distant metastases compared to their matched primary breast and prostate cancer specimens [39,40,41,42,43]. On top of that, metastatic tumors of invasive ductal adenocarcinoma origin express E-cadherin regardless of the E-cadherin status of the primary breast tumors [42]. Consistently, bone metastases from E-cadherin-negative, poorly differentiated breast carcinomas also exhibit E-cadherin expression [44]. All the above add one more layer of complexity regarding the role of cEMT in the cancer cell adaptation and establishment at the metastatic sites, instead pinpointing to E-cadherin as an essential player for the establishment and protection of metastatic deposits.

An abundance of markers have been associated with cEMP, including mesenchymal markers such as vimentin and N-cadherin, core cEMT-activating transcription factors (SNAIL1/2, ZEB1/2, TWIST1/2), and loss of epithelial protein expression (E-cadherin, EpCAM) (31). Of these, E-cadherin might be the strongest, since its loss is a hallmark of cEMT and strongly correlates with dissemination even in the absence of mesenchymal marker expression [27,45]. This is the basis for the partial epithelial and mesenchymal phenotypes noted in cancer cells.

The role of E-cadherin is extremely important in “shielding” micrometastases from chemotherapy as shown by Ma et al. in two studies in prostate cancer, where it was shown that hepatocytes induce a cMErT of prostate cancer cells via initial suppression of the MAP kinases p38 and ERK, conferring an epithelioid morphology to prostate cancer cells with an increase/re-expression of E-cadherin and other epithelial markers (ZO-1, connexin 43). Upon hepatocyte-dictated E-cadherin upregulation in prostate cancer cells and cell-cell adhesion through homeotypic binding, prostate cancer cells reactivated canonical survival pathways, such as PI3K/Akt, MEK/ERK, and JAK/STAT in response to chemotherapy and/or death signals, gaining resistance to treatment in a proliferation-independent manner [46,47] (Figure 1). The translational impact of these findings is impressive, as these pathways could be targeted in the context of combinatorial therapeutic approaches to synergistically improve the efficacy of traditional chemotherapies by using small doses of MEK inhibitors not for their direct therapeutic effect, but rather for re-sensitizing small micrometastases to existing chemotherapeutic agents. 

## 5. Disseminated Tumor Cell Dormancy

Upon successful extravasation of DTCs into the metastatic site, these either exist as individual cells in a state of quiescent dormancy or form small masses that fail to expand. This cellular dormancy represents one of the most important mechanisms driving the resistance of disseminated tumor cells to chemotherapy and potentially immunotherapy for reasons that will be presented next.

As circulating tumor cells arrive at the metastatic site and extravasate, the first thing they encounter is the basement membrane of the endothelium and its supporting stroma, comprising the perivascular niche. Stepping into the perivascular niche drives DTCs to enter into a state of quiescent dormancy and remain dormant for months or years until triggered to start proliferating again [48]. With the use of an in vitro engineered microvascular niche model recapitulating key elements of the pre-metastatic niche, Ghajar et al. showed that the proliferative fate of DTCs was determined by their proximity to blood vessels, strongly suggesting that blood vessel-derived cues promoted quiescence of adjacent DTCs [49]. Since most chemotherapeutic agents in clinical use target-specific components of the proliferative capacity of cancer cells, such as microtubule formation, response to cytotoxic therapies critically depends on cell cycle activity; the cell cycle arrest in dormant cells confers resistance to therapy [50]. This logical explanation was first supported by evidence arising from two studies measuring absolute numbers and proliferation indices (Ki-67) of DTCs in bone marrow before and after adjuvant therapy [51,52,53]. DTCs were mostly negative for proliferation markers; moreover, their survival post-chemotherapy was associated with later relapse. Therefore, relapses occurring after months or years are presumed to arise from DTCs lying dormant at least over the period of several months during and after drug administration. Nevertheless, subsequent studies by Carlson et al. expanded the concept of the protective role of the perivascular niche even to non-proliferating cells, as endothelial cells protected human basal breast cancer T4-2 cells from apoptosis induced by paclitaxel, doxorubicin, 5-FU, or lapatinib, and the chemotherapy-resistant cancer cells comprised both an mVenus/p27-positive quiescent and an mVenus/p27-negative proliferating population [54]. Furthermore, proteomic analysis of the perivascular niche identified integrin-mediated adhesion of cancer cells to the extracellular matrix (ECM) as the main mechanism of perivascular niche-mediated resistance to chemotherapy. Notably, while endothelial cells of mature microvessels deposited thrombospondin-1 (TSP-1) into the basement membrane and promoted dormancy of DTCs, areas of sprouting angiogenesis lacked TSP-1 expression, thus exerting an opposing effect to the dormant DTCs [49].

It is now widely accepted that the microenvironment influences DTC fate through a variety of different signals. Bragado et al. provided the first evidence of the bone marrow being a growth-restrictive “soil,” where TGFβ2 and p38 signaling induce DTC dormancy [55]. Master regulators of hematopoietic stem cell (HSC) dormancy in the bone marrow, including BMP4, BMP7, TGFβ2, and GAS6, could induce residual disease dormancy in many types of cancer, including breast and prostate cancer [56,57,58,59,60]. Importantly, many of the aforementioned signals induce ERK inhibition and p38 activation, and thus, a low ERK/p38 signaling ratio, which drives a G0/G1 arrest and entrance into dormancy [61]. From a mechanistic point of view, increased p38 kinase activity induces activation of the unfolded protein response (UPR), which regulates the endoplasmic reticulum stress-related transcription factor ATF6, in turn promoting survival of dormant cells via Rheb upregulation and mTOR signaling [62,63]. Consistently, induction of protein disulphide isomerase and GRP78 in dormant cells secondary to p38 activation results in increased survival of dormant cells when exposed to chemotherapy and nutrient deprivation via BAX inhibition, potentially through BIK upregulation [64]. 

Many studies have shown that systemic cancer spread potentially occurs long before diagnosis; thus, acquisition of genetic alterations for DTCs to drive metastasis frequently occurs within the metastatic site [6,8]. Therefore, the presence of a niche promoting acquisition of metastasis-supportive adaptations and providing protection against chemotherapy is beneficial for early DTCs. Direct measurement of local oxygen concentrations in bone marrow revealed hypoxic regions sufficient to trigger an NR2F1-regulated dormancy and chemoresistance program [65]. Although identified in primary tumor cells, hypoxic conditions in the bone marrow may induce dormancy in arriving DTCs as well [66]. Additionally, this NR2F1-driven program was found to induce expression of pluripotency genes in experimental models of DTCs [67]. Given that chemotherapy alters the local oxygen levels in the bone marrow [65], it can potentially enhance stemness traits in dormant DTCs fueling metastatic progression. Overall, adjacent cellular elements combined with non-cellular microenvironmental parameters (oxygenation, ECM) have a dramatic impact on the response to therapy, and therapy per se represents a part of the equation between emergence versus dormancy. 

## 6. MME and Metastatic Reactivation

The role of MME is fundamental not only for induction of dormancy in DTCs and micrometastases, but also for the successful micrometastatic reactivation and treatment resistance within this context as well. This emergence from dormancy relies not only on the immediately adjacent microenvironment (e.g., endothelial cells, stromal fibroblasts), but also systemic signals and ECM components governing several processes, including neoangiogenesis, inflammation, and fibrosis [68,69,70,71].

By co-culturing breast cancer cell lines MCF-7 and MDA-MB-231 with hepatic non-parenchymal cells (NPCs) and endothelial cells, Taylor et al. showed that NPCs in the metastatic hepatic niche secrete factors inducing a mesenchymal shift in these epithelial breast cancer cells initiating outgrowth partly via EGFR signaling activation [72]. Furthermore, macrophages are also important players regarding metastatic outgrowth as macrophage phenotypic subtypes M1-like and M2-like were shown to diametrically regulate cEMP in metastatic breast cancer cells, with M2-like macrophages conferring a mesenchymal phenotype to breast cancer cells imparting a growth advantage [73]. In agreement with the notion that specific signals from the MME promote micrometastatic activation via a bidirectional crosstalk, it has been shown that IL-11 secreted by breast cancer cells induces recruitment and activation of osteoclasts, which foster micrometastatic expansion in the bone [74]. Moreover, breast cancer cells promote osteolytic bone metastasis by producing Jagged1 and engaging Notch signaling in bone-resident cells [75].

The role of a dense, fibrotic MME is also essential for the emergence of macrometastases. Fibrosis with enrichment of collagen type I at the metastatic site can act as a critical determinant of cytoskeletal reorganization in dormant breast cancer cells, leading to their transition from dormancy to metastatic growth [76,77]. Additionally, collagen I-mediated signaling through discoidin domain receptor tyrosine kinase 1 (DDR1) is a requirement for the reactivation of dormant breast cancer cells [78]. Heavy crosslinking of collagen fibers, including those created by HIF1-induced lysil oxidase, can induce metastatic reactivation by promoting integrin-mediated conversion of mechanical forces to biochemical cues [76,79,80,81]. In a similar manner, cancer-associated fibroblasts (CAFs) and hepatic stellate cells (Ito cells) have been shown to produce ECM rich in collagen, fibronectin, and laminin, thus fostering a dense, fibrotic stroma, and subsequent binding of breast cancer cells to these components confers resistance to cytotoxic chemotherapy via apoptosis inhibition [82,83,84]. Finally, a recent study by Albrengues et al. showed that stimuli of inflammation, such as bacterial lipopolysaccharide (LPS) or cigarette smoke can activate neutrophils to release their DNA content into the lung parenchyma forming neutrophil extracellular traps (NETs), which are associated with neutrophil degranulation and release of neutrophil elastase and MMP9. While NETs act as scaffolds, these powerful proteases enhance the processing of laminin-111, a major basement membrane component, which subsequently binds α3β1 integrin and activates focal adhesion kinase (FAK) signaling in dormant DTCs. This phenomenon induced proliferation of DTCs in mouse models of cancer, while the use of an α3β1 integrin-blocking antibody prevented DTC reactivation in NET-rich lungs [85]. 

Overall, it becomes clear that the induction of a dense ECM rich in collagen, fibronectin and laminin in the MME, which promotes emergence from dormancy and a secondary cMErT, represents one major mechanism of resistance to chemotherapy (Figure 2). Given the association between quiescence and protection from chemotherapy, this finding might seem counterintuitive. Nevertheless, it is also true that other mechanisms are critical regulators of resistance, such as the binding of tumor cells to these ECM components activating downstream pathways and triggering inhibition of drug-induced apoptosis and conferring increased resistance even to targeted therapeutics [82,83,84]. On top of these, the matricellular protein tenascin C also provides for survival by tonic signaling via the EGFR [86]. Furthermore, the stiffness of the matrix contributes to treatment resistance by hindering the diffusion of chemotherapeutic agents into cancer tissues via increased hydrostatic pressure [87]. Based on the observation that tumors originating from “stiff tissue” (lung, skin, bone) have significantly higher number of somatic mutations and copy number alterations compared to tumors from ‘’soft tissues’’ (bone marrow, brain), it has been suggested that tumor stiffness could represent a major driver of genomic instability through increased cell proliferation (increasing the probability of spontaneous mutations), increased frequency of nuclear envelope rupture, and higher selective pressure [88]. Finally, the role of the immune component of the MME is detrimental but will be discussed in detail below in ‘’immune surveillance.’’ 

## 7. Immune Surveillance

Over the last years, it has become increasingly clear how important the role of the immune system is in controlling dormancy, mediating treatment resistance and triggering reactivation of metastatic cancer. This knowledge is driven not only by a multitude of mechanistic studies, but also by the striking success of checkpoint blockade immunotherapy with the use of CTLA-4- and PD-1/PD-L1-blocking antibodies [89]. The introduction in the clinic of immune checkpoint blockers has led to sustained long-term control of several types of cancer, including advanced melanoma, renal cell carcinoma, urothelial carcinoma, and non-small cell lung cancer, among others, though often only in small subsets of cancers. Nevertheless, outcomes are far from satisfactory with regards to other types of cancer, such as metastatic castration-resistant prostate cancer (mCRPC), and efforts are focused on combinatorial therapeutic interventions to improve efficacy.

The effects of the immune system on metastasis is a phenomenon starting early, with the immune cells implicated as critical regulators of all steps of the invasion-metastasis cascade [90,91]. Metastasis-associated macrophages (MAMs), especially those with an M2-like pro-tumorigenic phenotype, together with regulatory T lymphocytes (Tregs) and myeloid derived suppressor cells (MDSCs), aided by a number of cytokines and chemokines, foster a permissible premetastatic niche [92,93] (Figure 3). All three types of cells contribute to the formation of a protective microenvironment for the incoming DTCs by suppressing the cytotoxic functions of T and NK cells [94,95,96]. These cells are recruited and mobilized to play a pro-metastatic role by soluble factors and extracellular vesicles in the context of primary tumor-driven systemic processes that affect metastatic colonization [97]. In models of bladder cancer metastasis to the lung, it has been demonstrated that DTC-derived endothelin-1 (ET-1) and ET_A_R activity are necessary conditions for successful lung metastatic colonization, demonstrating not just immune modulation but also secretory support ensures from these cells. This is consistent with findings that macrophage infiltration of the tumor and the pulmonary parenchyma, resulting in secretion of trophic and immunosuppressive factors (MCP-1, IL-6), are also required for this process [98]. Similarly, in breast cancer metastasis models, monocytes (recruited to DTCs by CCL2 produced in the lung both by the tumor and the surrounding stroma) differentiate into MAMs that promote DTC proliferation [99,100,101,102]. Moreover, in a genetic mouse model of pancreatic ductal adenocarcinoma, it has been shown that granulin secretion by MAMs induced transformation of hepatic stellate cells into myofibroblasts, which in turn secreted periostin and induced the formation of a fibrotic microenvironment, favorable for activation of integrin signaling [103].

Successful eradication of DTCs is dependent on the effector Ly6C+CD8+ T cell population and the inability of a tumor to induce a suppressive granulocytic MDSC population [104]. The presence of MDSCs in metastatic lesions correlate with several negative parameters, including advanced disease stage, poor responses to chemotherapy, as well as resistance to immune checkpoint blockade [105,106,107,108,109]. The importance of MDSCs is further illustrated by recent work on mCRPC. Signaling mediated by MDSC-secreted IL-23 and its receptor on prostate cancer cells promotes the development of mCRPC, and this dismal phenomenon can be delayed by the use of pharmacologic agents blocking IL-23-mediated signaling [110]. Additionally, MDSCs are critical in generating a suppressive tumor microenvironment by induction of PD-L1, iNOS, arginase-1, IDO, and osteopontin, thus inhibiting T cell and NK cell tumoricidal activities and stimulating Treg cells [95,111,112,113,114].

The immune response is dynamic and tightly regulated, in the sense that signals that enhance anti-tumor immune responses concurrently turn on inhibitory genes that function as “immune breaks.” For example, initial T cell activation, via T-cell receptor signaling and CD28 co-stimulation, eventually leads to increased expression of the immune checkpoint inhibitor, CTLA-4 [115]. Similarly, effector Th1 responses, such as increased IFNγ production, concurrently lead to increased expression of PD-L1 on multiple cell types, including tumor cells and macrophages, which can engage the PD-1 receptor on T cells to suppress anti-tumor immunity [116,117]. As a consequence, metastases frequently exploit the aforementioned mechanisms to escape immune surveillance, and even the perivascular niche itself can suppress immune responses through expression of checkpoint ligands, such as PD-L1 [118]. Recently, combinatorial immunotherapy of immune checkpoint blockade, anti-CTLA-4 or anti-PD-1 antibodies, with multikinase inhibitors targeting MDSCs (cabozatinib and BEZ235) showed robust synergistic response in preclinical models of mCRPC [119]. 

An interesting aspect relevant to DTCs evading immune detection is related to antigen processing and presentation through major histocompatibility complex class I (MHC-I) molecules. Decreased expression of MHC-I in DTCs was initially suggested based on studies of breast, stomach, and colon carcinoma [52,53]. On top of this, breast and lung cancer DTCs were reported to broadly downregulate ligands for an immune system-activating receptor on NK cells, named ULBP1, thus achieving evasion of NK-cell-mediated clearance. By expressing a Sox-dependent stem-like state and inhibiting WNT signaling in an autocrine manner, DTCs were able to enter quiescence and evade innate immunity to remain latent for extended periods [120]. Recently, Pommier et al. showed that dormant DTCs activate components of the unfolded protein response (UPR), causing posttranscriptional downregulation MHC-I molecules, thus suppressing antigen presentation to CD8+ T cells [121]. This is in line with evidence from quiescent hair follicle stem cells, which, although non-neoplastic, reside in specialized niches that control their quiescence, and in this state downregulate MHC-I and evade CD8+ T cell immune recognition [122].

The ability of metastatic deposits to evade immune detection is notoriously evident in the case of outgrowing metastases. Immunosuppressive cancer microenvironments are now recognized as a major hurdle for positive clinical outcomes regarding the use of immune checkpoint blockade. Some immunotherapy-resistant tumors are characterized by a paucity of neoantigens that are available for presentation (reflecting a low tumor mutational burden), and ultimately, this presents an obstacle for antigen recognition [123,124]. This difficulty is particularly evident in mCRPC, a traditionally “immunocold” tumor, where multiple clinical trials of immune checkpoint inhibitors failed to produce improvement in progression-free survival or overall survival. In other cases, despite expression of PD-1 and PD-L1, failure of immune checkpoint inhibitors can be explained by somatic JAK1/2 mutations rendering tumor cells insensitive to IFNγ (this can be either a primary or acquired resistance mechanism) or loss of β2-microglobulin impairing tumoral antigen presentation [125,126]. Notably, some innately resistant metastatic tumors have been shown to display a transcriptional signature referred to as IPRES (innate anti-PD-1 resistance), indicating concurrent upregulation of genes involved in mesenchymal transition, cell adhesion, extracellular matrix remodeling, angiogenesis, and wound healing [127].

## 8. Epigenetic Modifications in Metastasis

Epigenetic mechanisms of transcriptional regulation include DNA methylation, histone modifications, regulation by non-coding RNAs, and the dynamic binding of various proteins shaping the chromatin compaction status, thus spatially regulating the accessibility of chromatin to transcriptional factors and the ability to form active transcriptional higher order chromatin organization [128,129]. DNA methylation represents the most widely studied epigenetic modification and usually refers to the covalent modification of cytosine at the 5ʹ position at sites preceding guanine (CpG) [130]. Epigenetic modifications are increasingly recognized as a major contributor to the development of treatment resistance both to targeted agents, but also chemo- and radiotherapy in general, and can even predict treatment response [131].

Phenotypic differences between tumor cells demonstrating stemness versus those lacking it can be predominantly attributed to epigenetic changes within the context of cEMT in the former, and this is a critical attribute for establishment of treatment resistance [132]. One particularly relevant target of epigenetic regulation is that of E-cadherin downregulation, the sine qua non of cEMT; this target is suppressed at different levels ranging from autocrine signaling-induced internalization and transcriptional repression (such as in prostate cancer) to DNA methylation (as in breast cancer) [27]. The value of such epigenetic downregulation rather than genetic mutation or deletion is the reversibility/phenotypic plasticity needed for metastatic seeding [26]. The presence of E-cadherin expression provides for a pan-resistance to chemotherapies due to the downstream survival signals as discussed above.

It has also been proposed that open chromatin structure incorporates traditional chemotherapeutics, such as DNA-damaging agents carboplatin and doxorubicin, more easily than compact chromatin, and similar correlations apply to irradiation [133,134]. Within this context, epigenetic targeting drugs have been shown to increase accessibility of DNA-damaging agents via global chromatin decompaction, and consequently, trigger DNA damage and apoptosis in breast cancer models [135]. With reference to the predictive value of epigenetic changes in treatment response, the most representative example is MGMT (O-6-methylguanine DNA methyltransferase) hypermethylation in gliomas, which represents the best independent predictor of response to carmustine and temozolomide, since MGMT hypermethylation renders tumor cells more sensitive to treatments and is associated with regression of tumor and prolonged overall survival [136,137].

Finally, the role of epigenetics in defining responses to immunotherapy is an area of tremendous interest. Strikingly, melanoma with low PD-L1 expression and global hypermethylation represents a large group of patients exhibiting primary resistance to immune checkpoint blockade therapy and this hypermethylation is associated with overexpression of DNA methyltransferases (DNMTs) and histone methyltransferase EZH2 in the PRC2 repressive complex [138]. Consistently, epigenetic therapies (such as DNMT inhibitors) have been increasingly reported to enhance antigen processing and presentation pathways in tumors, especially by upregulating MHC-I, therefore potentiating responses to immunotherapies [139,140,141,142,143,144]. In the years to come, it will be particularly interesting to see the results from clinical trials combining epigenetic therapies with immune checkpoint blockade especially in the case of traditionally immune “cold” cancers, such as prostate cancer.

## 9. Metabolic Rewiring and Metastasis

All the steps of the metastatic cascade, from escape of tumor cells from the primary tumor up to emergence of lethal macrometastases, are mirrored by tumor metabolism and metabolic rewiring. It is also clear that survival in the metastatic environment imposes metabolic requirements distinct from those supporting cell growth; inhibition of these adaptive activities results in decreased metastatic spread [145]. Moreover, it is becoming increasingly evident that this metabolic plasticity contributes significantly to the primary (and acquired) therapeutic resistance of metastatic cancer [146]. 

As noted above, the metastatic cascade is a highly inefficient process characterized by the presence of multiple bottlenecks, but metabolic reprogramming enables successful navigation through some of these barriers [145]. Initial steps of local invasion and intravasation require release of CO2, lactate, and other organic acids from metabolically active tumor cells promoting acidification of tumor microenvironment, decrease in the number of adherens junctions between tumor cells, and degradation of ECM [147]. Other metabolic alterations support cEMT which is also essential for this step. One of them is the oncogene-dependent activation of UDP-Glucose-6-Dehydrogenase (UGDH) resulting in the depletion of UDP stores and enhancing the expression of SNAIL [148]. The other one is expression of asparagine synthetase (AsnS), involved in the synthesis of the conditionally essential amino acid asparagine, which allows for breast cancer cell invasiveness and metastasis as there is a disproportionately high asparagine content of cEMT-associated proteins [149]. Consequently, blood-borne dissemination requires mechanisms for adequate NADPH and glutathione production to overcome the temporary oxidative stress, as shown in melanoma patient-derived xenograft models and genetically engineered mouse models of melanoma, breast cancer, and lung cancer [150,151,152,153,154]. Nevertheless, this is likely a cancer-specific phenomenon, since other models support that some tumor cells are not susceptible to oxidative stress, and instead, reactive oxygen species (ROS) may promote metastasis in those models [155,156,157]. This point is convoluted by the short transit times in the circulation being outweighed by longer periods prior to successful integration into the ectopic host parenchyma. 

Colonization of distant organs is supported by metabolic parameters as well, and potentially, the organotropism of several cancers is dictated not only by the lympho-vascular access to the respective organs, but also the balance between the metabolic requirements of tumor cells and the local milieu. Within this context, in breast cancer metastatic models to the lung, collagen crosslinking by the enzyme prolyl-4-hydroxylase is a process heavily dependent on a-ketoglutarate produced by the very tumor cells [158]. Interestingly, some cancers exhibit a strong preference for fatty acids as an energy fuel. Prevention of fatty acid transfer from adjacent adipocytes to ovarian carcinoma cells has been shown to reduce disseminated growth in mice [159]. In a similar fashion, triple-negative breast cancer may rely on fatty acid oxidation for sustenance of aberrant Src activity promoting metastasis [160].

The metabolic rewiring of cancer cells not only supports successful metastatic seeding but also contributes to therapeutic resistance in the metastatic setting against chemo-, radio-, and immunotherapy [146,161]. Intensive aerobic glycolysis and lactate production induces immunosuppression and therapy resistance [162]. Inhibitors of several glycolytic enzymes and glucose transporters are currently being tested in preclinical or clinical studies to counteract resistance to chemotherapeutic agents [146]. Lipid metabolism is a major determinant of therapeutic resistance, with fatty acid synthase (FASN) overexpression inducing resistance to various drugs, including adriamycin and mitoxantrone, in breast cancer cells [163] and radiotherapy-resistant head and neck squamous cell carcinomas [164]. Dysregulated sphingolipid metabolism may be another key contributor to therapeutic resistance as shown in studies of ovarian cancer [165]. Sequencing studies in human melanoma patient samples showed that melanoma brain metastases are enriched for gene sets related to oxidative phosphorylation (OXPHOS), and the use of an OXPHOS inhibitor improved survival of mice bearing MAPK inhibitor-resistant intracranial melanoma xenografts [166]. Finally, caveolin 1, the main component of the caveolae plasma membranes found in most cells, has been associated with altered metabolism in prostate cancer and subsequent radiologic and multidrug resistance [167,168].

In sum, the metabolic alterations accompany both metastatic spread and generalized resistance to a wide range of therapies. Studies in animals support a causal relationship between the reprogramming and outcomes. Human studies or adjuvant therapies will be needed to determine whether these metabolic changes are drivers and adaptations to ectopic seeding.

## 10. Further Mechanisms of Resistance 

There is a multitude of other mechanisms mediating primary, intrinsic pan-therapeutic resistance of metastatic deposits, especially with reference to larger, outgrowing metastases. Cancer cells can form spatial gradients that perturb adequate blood flow, thus creating a pro-tumorigenic hypoxic environment simultaneously reducing the exposure of the tumor to chemotherapeutic drugs [169]. This might partially explain the efficacy of certain anti-angiogenic agents, since they not only attenuate blood flow to tumors, but also normalize vascular structure and function to improve delivery of systemic agents [170,171]. Relevant examples include the synergistic activity exhibited by the combination of anti-PD-1/PD-L1 antibodies with anti-angiogenic tyrosine kinase inhibitors (lenvatinib, axitinib) in recent clinical trials of advanced endometrial and renal cancer, respectively [172,173].

Last but not least, a major driver of primary, intrinsic therapeutic resistance, particularly in emergent macrometastases, is intratumoral heterogeneity. As DTCs or minute micrometastases emerge from dormancy, cancer cells start proliferating, and thus, progressively acquiring new genomic alterations through a variety of mutational processes with different evolutionary speeds, from the slow rate of age-related mutations to frequent gene editing by APOBEC enzymes, to catastrophic events due to genomic instability, chromothripsis, and chromosomal instability [174,175,176,177]. Exogenous exposures and internal environmental dynamics pose ecosystem-selective pressures ultimately leading to parallel and convergent evolution with spatial segregation of different clones and subclones, and this intratumoral heterogeneity can represent another layer of complexity to primary treatment resistance exhibited by metastatic cells [178]. Of course, cancer therapies themselves might represent the most crucial evolutionary pressure, but this adaptive response is not relevant to primary intrinsic resistance mechanisms reviewed here. Still, the experience that micrometastases and emergent nodules present as resistant to multiple chemotherapies suggest that therapy-specific mutational resistance is secondary to the MME-imprinted generalized resistance.

## 11. Conclusions

Resistance of metastatic cancer to therapy remains the most challenging problem in oncology today, and the primary, intrinsic pan-resistance exhibited by DTCs, micrometastases, and even macrometastases represents only one side of this coin. Given that each tumor has its own defining set of genomic and molecular characteristics, and that dormancy can take different forms unique to each type of cancer, solving the problem of therapeutic resistance remains challenging. Moreover, the bulk of techniques utilized to approach these phenomena thus far represent static studies (“snapshots”) providing only endpoint measurements in vivo. Even dynamic tools such as intravital imaging have major limitations since it is currently almost impossible to track individual dormant cells or small groups of cells in animals. Efforts to capture spatial and temporal heterogeneity of metastases and the microenvironment require large numbers of experimental animals making the cost prohibitive for most laboratories.

Here, we provide a framework that dissects therapeutic pan-resistance into its biological determinants. This perspective enables different problems to be tackled separately and then integrated in a global manner. It is clear that the role of the MME is essential either by directly determining entry and exit from dormancy or by providing a “fertile” soil for acquisition of mutations to drive resistance and outgrowth. Furthermore, our steadily increasing understanding of the role of different components of the immune system in keeping these metastases in check or causing their outgrowth provides further insight and additional translational opportunities. The conceptual roadmap for tackling the problem of resistance is set, and as more sophisticated molecular biology techniques, imaging modalities, single-cell sequencing technologies, bioinformatics, and precision medicine advance, their amalgamation will advance the field in unprecedented ways. Overall, understanding cellular dormancy, the role of the MME, and the immune system is required in order to upgrade our therapeutic armamentarium against metastatic cancer and write the next chapter in the history of medicine.

## Figures and Tables

**Figure 1 ijms-21-07304-f001:**
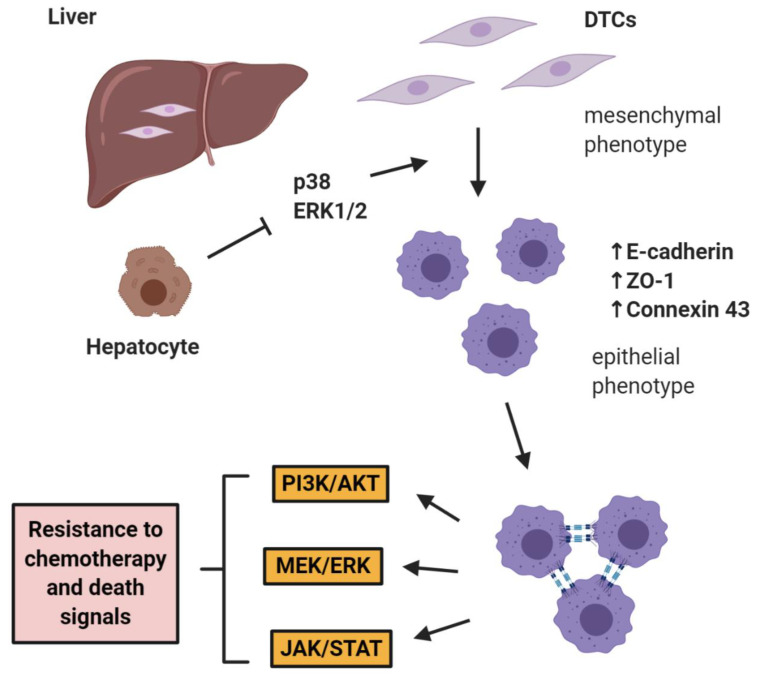
E-cadherin mediates micrometastatic dormancy and chemoresistance. Hepatocytes induce E-cadherin upregulation in metastatic prostate cancer as well as other epithelial markers, thus promoting a partial cancer-associated mesenchymal-to-epithelial reverting transition (cMErT) by initial suppression of p38 and ERK. Upon cell-cell E-cadherin ligandation, stimulation of metastatic cells with chemotherapy induces activation of canonical pro-survival kinases resulting in increased chemoresistance. This chemoresistance is proliferation-independent and can be targeted with an adjuvant treatment to achieve optimal efficacy with traditional chemotherapies. DTCs: disseminated tumor cells.

**Figure 2 ijms-21-07304-f002:**
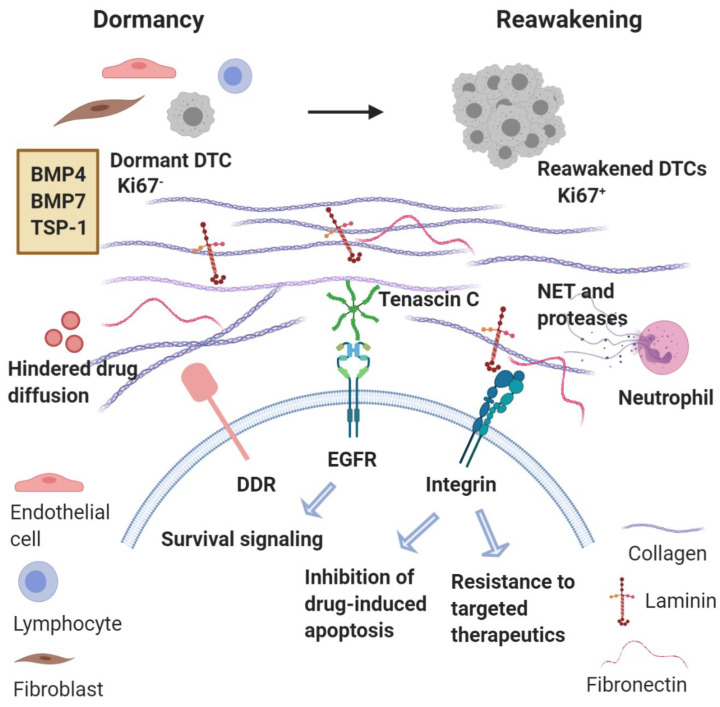
Metastatic microenvironment dictates emergence from dormancy and treatment resistance. Reawakening of disseminated tumor cells (DTCs) from dormancy may be triggered by changes in the microenvironment in which these DTCs reside or even changes in immune activity. Key signaling components involved in metastatic reactivation include collagen, laminin, and fibronectin interacting with integrins and DDR (Discoidin Domain Receptor). Neutrophils recruited can produce neutrophil extracellular traps (NETs) and the associated proteases permit the integrin-mediated activation of dormant DTCs that proliferate to form macrometastases. Concurrently, binding of tumor cells to these ECM components trigger inhibition of drug-induced apoptosis and resistance to targeted therapies. Tonic activation of EGFR by tenascin C also provides survival signaling. Finally, the density of the matrix impedes the diffusion of chemotherapy agents.

**Figure 3 ijms-21-07304-f003:**
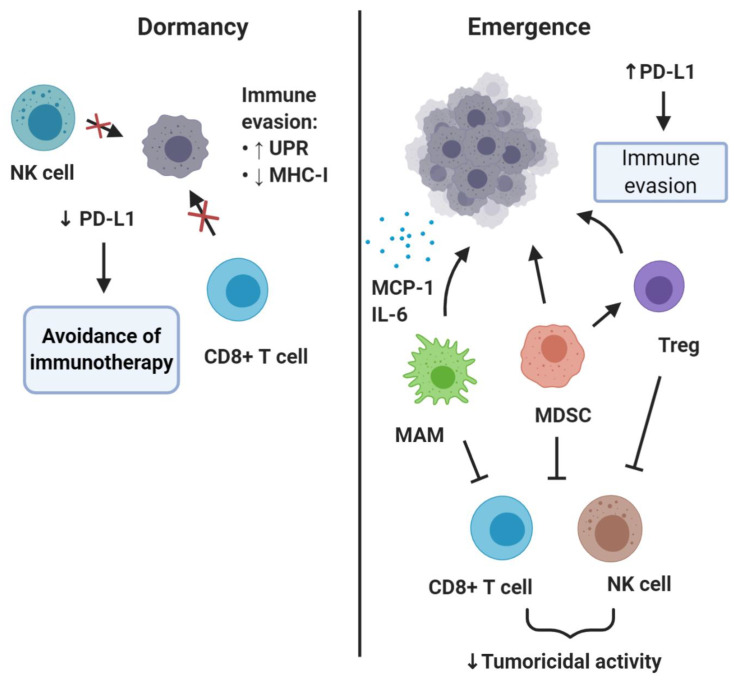
Roles of immune system in DTC dormancy and reawakening. This simplified schematic diagram shows that dormant DTCs may upregulate unfolded protein response (UPR) activity, which results in MHC-I downregulation and thus impaired immunological visibility to NK cells and CD8+ (cytotoxic) T cells. Downregulation of PD-L1 is a potential mechanism through which dormant DTCs may evade immunotherapy based on PD-1/PD-L1-blockade. Emergence of outgrowing metastases is linked to the activities of several immune cells, including regulatory T cells (Treg), myeloid-derived suppressor cells (MDSC), and metastasis-associated macrophages (MAMs), and also upregulation of immune markers such as PD-L1.

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
