# Peer review of "A Perspective on Therapeutic Pan-Resistance in Metastatic Cancer"

_ijms, 2020, doi:10.3390/ijms21197304_

Round 1
Reviewer 1 Report
This review gives an interesting and comprehensive overview of the biology behind the clinical phenomenon of generalized resistance in metastatic disease. The manuscript is well written and reflects the actual state of research in this area including different aspects which influence the reactivation of metastatic growth of dormant, chemoresistant tumor cells. In this context, processes involved in metastasis e.g. the cancer-associated epithelial-mesenchymal plasticity (cEMP) are discussed as well as mechanisms that drive the resistance of disseminated tumor cells like cellular dormancy and the host-tumor crosstalk in the metastatic microenvironment.
The only concerns I have are about the novelty and originality of the present review, since the authors published a largely similar article in terms of content this year in Seminars in Cancer Biology (Bo Ma, Alan Wells, Amanda M. Clark: The pan-therapeutic resistance of disseminated tumor cells: Role of phenotypic plasticity and the metastatic microenvironment. Seminars in Cancer Biology 60 (2020): 138-147.). In my opinion, this article in the present form brings no significant added value compared to the previously published one.
Author Response
We thank the reviewer for generous comments on the manuscript. Although there is some overlap unavoidably with the mentioned article, the present work dives deeper in the concepts of cancer-associated epithelial-mesenchymal plasticity, disseminated tumor cell dormancy, and role of myeloid-derived suppressor cells and immune checkpoints in escape from immune surveillance. Moreover, it covers recent novel research findings on the significance of unfolded protein response and neutrophil extracellular traps for dormancy, immune evasion and metastatic reactivation. Finally, we have now included two detailed sections on the role of epigenetic and metabolic factors in therapy resistance.
Reviewer 2 Report
In the present review manuscript, the authors have reviewed the integrative molecular mechanism of therapeutic resistance in different types of metastatic cancer. The authors have well described the role of oncogenic signaling as well as inflammatory cytokines in therapy resistance. It will be good if authors can include the role of epigenetic and metabolic factors in therapy resistance.
Author Response
Thank you for your generous comments on the manuscript. We have now corrected the omissions that you pointed by an addition of two additional sections on the role of epigenetic (section 8) and metabolic (section 9) factors in therapeutic panresistance of metastatic cancer.
Round 2
Reviewer 1 Report
The authors significantly improved the quality of the manuscript by including epigenetic and metabolic aspects of metastasis and therapy resistance.